

# Collective oscillations of a two-component Fermi gas on the repulsive branch

Tomasz Karpiuk[1], Piotr T. Grochowski[2*],
Mirosław Brewczyk [1] and Kazimierz Rzążewski[2]

**1** Wydział Fizyki, Uniwersytet w Białymstoku,
ul. K. Ciołkowskiego 1L, 15-245 Białystok, Poland
**2** Center for Theoretical Physics, Polish Academy of Sciences,
Aleja Lotników 32/46, 02-668 Warsaw, Poland

⋆ piotr@cft.edu.pl

## Abstract

We calculate frequencies of collective oscillations of two-component Fermi gas that is kept on the repulsive branch of its energy spectrum. Not only is a paramagnetic phase explored, but also a ferromagnetically separated one. Both in-, and out-of-phase perturbations are investigated, showing contributions from various gas excitations. Additionally, we compare results coming from both time-dependent Hartree-Fock and density-functional approaches.



# 1 Introduction

Theoretical and experimental studies of multi-component quantum mixtures have always played a crucial role in our attempts to understand the quantum theory [1, 2]. For several decades, the central point of such considerations was reserved for mixtures of different states of helium [3]. However, the first realizations of quantum degenerate gases in 1990s [4–6] have created a playground for scientists that not only allows one to combine different fermionic and bosonic ingredients of a mixture, but also to freely tune interatomic interactions by means of Feshbach resonances [7].

One of the main techniques to study excitations of a trapped, interacting quantum mixture is to examine its collective oscillations [8, 9]. Depending of a particular scenario of an experimental excitation procedure, such oscillations can be classified into different categories [2]. Firstly, if a sample does not undergo a change in its volume, but its geometric shape is altered, the mode is called a surface one. Such modes were employed to study transitions from collisionless to hydrodynamic regimes in both bosonic [10,11] and fermionic [12,13] gases. On the other hand, if a volume change happens, one deals with a so called breathing or compression mode [14,15]. Those types of excitations have proved useful in investigations of equations of state for strongly interacting fermionic gases [16–18].

Both theoretical and experimental considerations [19–32] showed that collective oscillations in multi-component mixtures are affected by a variety of different effects, among the others, damping and frequency shifts [33–35]. Recently, much of the focus was centered onto center-of-mass oscillations (or spin-dipole) that have been employed to study coupling effects in mixtures of distinctive superfluids [36–39] and fermion-mediated bosonic interactions [40].

The richness of effects increases even more when decays through different channels and collapses [41, 42] or phase separations [43–46] come into play. Recently, oscillations in repulsive Fermi-Fermi [47, 48] and Bose-Fermi [49] mixtures in presence of phase-separating domain wall were studied experimentally. We focus on the former case, in which phase separation can be interpreted as a manifestation of the long-standing problem of ferromagnetic (Stoner) instability [47, 48, 50–60].

In such a system, two equally populated spin species (or in experimental terms – two hyperfine states) of Fermi gas interact only via repulsive short-range potential. However, the ground state of such a mixture is a superfluid of paired atoms of opposite spins (so called lower or attractive branch of the energy spectrum), rather than a ferromagnet (upper or repulsive branch) [57]. It stems from the fact that true zero-range interactions tuned by Feshbach resonances necessarily need an underlying attractive potential with a weakly bound molecular state [7].

Due to phase-separated state being intrinsically unstable towards decay into such molecules, early experiments dealt with high rates of losses [51]. To overcome these problems, in recent experiments artificial domain structure was initially prepared, making the phase-separated state stable for a finite time [47, 48].

As for now, theoretical studies have skipped analysis of small oscillations in this system while excited onto repulsive branch, instead focusing mainly on the attractive one [33,61–64]. In this work, we consider in- and out-of-phase radial and breathing modes of purely repulsive two-component fermionic mixture for both overlapping and phase-separated states. We compare our results with previously available weakly-interacting ones and show how to refine them by means of renormalization of the interaction. Moreover, we compare the results obtained by means of time-dependent Hartree-Fock calculations with the hydrodynamic approach [65], pointing out the regimes of applicability of the latter method.

The paper is structured as follows. In Sec. 2 we analyze excitation frequencies by means of time-dependent Hartree-Fock methods and compare them to past theoretical techniques.

Sec. 3 is dedicated to analysis of applicability of hydrodynamic methods in similar cases by comparing results to atomic orbital calculations. In the final Sec. 4, we briefly review obtained results and provide suggestions for further work.

## 2    Time-dependent Hartree-Fock calculations

We start our considerations by describing the method used to analyze the statics and dynamics of the mixture. We employ time-dependent Hartree-Fock (or atomic-orbital) approach [65, 66] in which we assume that many-body wave function $\Psi(\mathbf{x}_{1,+}, ..., \mathbf{x}_{N_+,+}, \mathbf{x}_{1,-}, ..., \mathbf{x}_{N_-,-})$ of $N = 2N_+ = 2N_-$ fermionic atoms in a two-component mixture is given by a product of single Slater determinants:

$$
\Psi = \frac{1}{\left(\frac{N}{2}\right)!}
\begin{vmatrix}
\varphi_{1,+}(\mathbf{x}_{1,+}) & . & . & \varphi_{1,+}(\mathbf{x}_{N_+,+}) \\
. & & & . \\
. & & & . \\
\varphi_{N_+,+}(\mathbf{x}_{1,+}) & . & . & \varphi_{N_+,+}(\mathbf{x}_{N_+,+})
\end{vmatrix}
\begin{vmatrix}
\varphi_{1,-}(\mathbf{x}_{1,-}) & . & . & \varphi_{1,-}(\mathbf{x}_{N_-,-}) \\
. & & & . \\
. & & & . \\
\varphi_{N_-,-}(\mathbf{x}_{1,-}) & . & . & \varphi_{N_-,-}(\mathbf{x}_{N_-,-})
\end{vmatrix}.
$$

(1)

The coordinates $\mathbf{x}_i$ of an atom denote spatial variables and $\varphi_{i,\pm}(\mathbf{x})$, $i = 1, ..., N/2$ denote different, orthonormal orbitals.

We restrict ourselves to contact interaction between different species, as such a description is accurate for systems with realistic short-range potentials (e.g. coming from broad Feshbach resonances). Such an interaction can be described by a single parameter, $s$-wave scattering length $a$, which can be connected to the coupling constant $g \geq 0$ by $g = 4\pi a\hbar^2/m$. Therefore, the time-dependent Hartree-Fock equations for the orbitals are given by

$$
i\hbar\partial_t \varphi_{i,\pm}(\mathbf{r}, t) = \left( -\frac{\hbar^2}{2m}\nabla^2 + V_{tr}(\mathbf{r}) + g n_{\mp}(\mathbf{r}, t) \right) \varphi_{i,\pm}(\mathbf{r}, t),
$$

(2)

for $i = 1, ..., N/2$ and with

$$
n_{\pm}(\mathbf{r}, t) = \sum_{j=1}^{N/2} |\varphi_{j,\pm}(\mathbf{r}, t)|^2.
$$

(3)

Here, $m$ is the mass of fermionic atom, $V_{tr}(\mathbf{r})$ is the trapping potential taken to be a spherically symmetric harmonic trap, $V_{tr}(\mathbf{r}) = \frac{1}{2}m\omega^2 r^2$, and $n_{\pm}$ are single-particle densities of both species.

To further refine the interspecies interaction and include many-body quantum corrections, we locally renormalize the coupling strength by replacing the bare scattering length, $a$, by the effective (and symmetrized) one:

$$
a_{eff} = [\zeta(k_+ a)/k_+ + \zeta(k_- a)/k_-]/2,
$$

(4)

where $k_+(\mathbf{r})$ and $k_-(\mathbf{r})$ are local Fermi momenta for the first and the second component, respectively, and $\zeta(\tilde{k}_F a)$ is a renormalization function [65, 67]. The renormalization function $\zeta(\tilde{k}_F a)$ is expanded perturbatively into powers of $\tilde{k}_F a$:

$$
\zeta(\tilde{k}_F a) = \tilde{k}_F a + B(\tilde{k}_F a)^2 + D(\tilde{k}_F a)^3 + ....
$$

(5)

For the physical interpretation and numerical values of consecutive perturbative terms, see e.g. [65]. Taking only first-order terms in the above expansion results in mean-field Eqs. (2).

However, taking into account the second- and third-order terms changes the time-dependent Hartree-Fock Eqs. (2) in the following way:

$$g n_\pm \rightarrow g n_\pm + C(4/3\, n_\mp^{1/3}\, n_\pm + n_\pm^{4/3}) +$$
$$E(5/3\, n_\mp^{2/3}\, n_\pm + n_\pm^{5/3}), \tag{6}$$

where $C = g a B (6\pi^2)^{1/3}/2$, $E = g a^2 D (6\pi^2)^{2/3}/2$.

The HF equations can describe both statics and dynamics of the mixture – the usual method is employment of the split-step operator to propagate the equations in time [66]. To obtain a ground state, this time is imaginary, and for the dynamics – real. We solve the coupled nonlinear partial differential equations (2) with the self-implemented solver prescribed in Ref. [66]. The size of the grid is $m_x = m_y = m_z = 128$ points and the time step is $\Delta t = 0.0005$ in oscillator units.

## 2.1 Initial states – imaginary time propagation

The usual number of atoms we use in our calculations is $N = 56 + 56$. It is far below the experimental values, however the behavior of the considered mixture is universal[1] when described in terms of dimensionless interaction parameter, $k_F a$. $k_F$ is the Fermi wave number in the center of a trap in Thomas-Fermi approximation, $k_F = (24N)^{1/6}/a_{HO}$, and $a_{HO}$ is the harmonic oscillator length. The number of atoms we consider proves to be more than enough to be in this universal regime. Moreover, we confirm this assumption by performing single calculations with larger numbers of atoms.

In Fig. 1 we present the results of propagation of the HF equations in imaginary time, yielding ground states of the mixture for given interactions. We can see that for small interactions, the clouds fully overlap (gas stays fully unpolarized), but become larger as the interaction increases. At some critical interaction strength, $k_F a_c$, phase separation occurs. For the interaction only slightly above the critical one, this separation happens only in the middle of the trap – the gas on the perimeter of the cloud stays unpolarized. What is more, the surface at which the separation occurs differs from one numerical realization to another – there is no privileged direction in which the splitting can occur. To ensure that the domain wall manifests perpendicularly to the $z$ axis, we introduce very small additional linear potential in $z$ direction that assists with the symmetry breaking. This linear potential is of opposite direction for different species.

For a very strong interaction, there is no longer any appreciable unpolarized gas on the perimeter and the clouds have vanishingly small overlap. In Fig. 1 each of the scenarios is presented with a single particle density along the $z$ axis. The single density profile in other directions preserves cylindrical symmetry.

In order to get value of $k_F a_c$ consistent with both more sophisticated theoretical approaches (Quantum Monte Carlo, LOCV, large $N$ expansion) and experimental results, we renormalize perturbatively the scattering length in the way described above. In the trap, this critical value is $k_F a_c \approx 0.9$.

## 2.2 Monopole compression mode

We start with the case of monopole compression mode. It is usually performed by symmetrically perturbing both clouds, by either slightly expanding or squeezing the trap, exciting oscillations with radial symmetry preserved. The gas changes its volume, therefore this mode

---

[1]The word *universal* relates to the behavior of the observables we consider – the collective frequencies and the interaction strength at which the phase separation happens; On the other hand, e.g. density profiles differ, slowly approaching Thomas-Fermi approximation with the growing number of atoms.

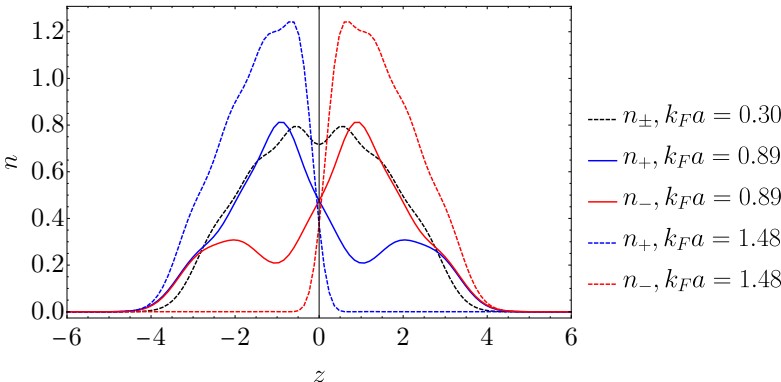

Figure 1: Ground-state densities for different values of interaction strength. For weak interaction, clouds stay unpolarized, but for some critical value of repulsion, they enter the ferromagnetic phase. Initially, just above the transition, gas becomes partially polarized only in the center of the trap, being unpolarized on the perimeter. For strong enough interaction, the overlapping part vanishes, leaving the clouds fully separated.

is classified as a compression one. The frequency of oscillations is taken from analysis of the cloud's width in a radial or axial direction.

Firstly, we compare previous theoretical approaches in a weakly interacting regime without a renormalization of the interaction strength. From here onwards, we will use atomic units, $\hbar = \omega = m = 1$. The starting point is a sum rule approach, firstly used for the oscillations in spin half Fermi gas by Vichi and Stringari [33]. The expression for a monopole mode in a harmonic trap reads

$$\omega_R = \sqrt{3 + \frac{E_{\text{kin}} + 3E_{\text{int}}}{E_{\text{tr}}}}. \tag{7}$$

We are interested in a weakly interacting case, for which phase separation does not occur, and Thomas-Fermi ground-state densities provide a reliable approximation. Under the assumption that spherical symmetry is not broken and both clouds fully overlap, $n(\mathbf{r}) = n(r) = n_+(r) = n_-(r)$, consecutive mean field energies read

$$E_{\text{kin}} = \frac{2^{8/3} 3^{5/3} \pi^{7/3}}{5} \int n^{5/3}(r) r^2 \mathrm{d}r,$$

$$E_{\text{int}} = 4\pi g \int n^2(r) r^2 \mathrm{d}r,$$

$$E_{\text{ho}} = 4\pi \int n(r) r^4 \mathrm{d}r. \tag{8}$$

In Ref. [33], noninteracting Thomas-Fermi profiles

$$n(r) = \left( \frac{(3N)^{1/3} - \frac{1}{2} r^2}{A} \right)^{3/2} \tag{9}$$

are used, where $A = \frac{6^{5/3} \pi^{4/3}}{12}$. The result is then quite easily calculated analytically, yielding simple expression

$$\omega_R \approx 2\sqrt{1 + 0.11 k_F a}. \tag{10}$$

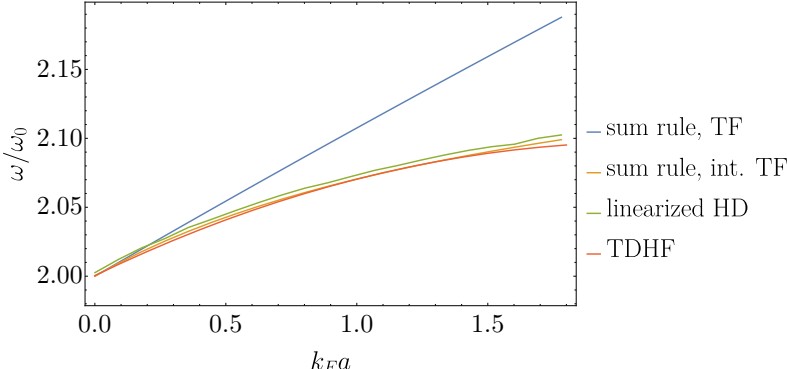

Figure 2: Frequency of a monopole mode calculated with the help of four different methods within the nonrenormalized mean field regime: sum rule approach from Ref. [33] using noninteracting and interacting Thomas-Fermi ground-state density profiles, linearized hydrodynamic equations and time-dependent Hartree-Fock equations. We can see that the latter three give very similar results, while the first on greatly overestimates the frequencies.

This result is shown in Fig. 2 as a blue line.

However, one can refine this result by considering ground-states profiles that come from Thomas-Fermi equations with mean-field interactions included (see Eq. 12 in Ref. [59]):

$$An^{2/3} + gn = \mu - \frac{1}{2}r^2, \tag{11}$$

where $\mu$ is a global chemical potential, that is implicitly given by the number of atoms through normalization procedure. This equation can be readily solved (see Appendix in Ref. [59]) by means of e.g. Cardano formulae, yielding ground state profiles that have analytical, however quite complicated form. Using these density profiles, one gets an orange line in Fig. 2.

Apart from curves from sum rule approach, there are two more results presented in Fig. 2: oscillation frequencies extracted from analysis of time evolution of TDHF equations, and frequencies from linearization of hydrodynamic formulation of the problem (further discussed in Sec. 3). One can see that all of them give very similar results for this particular mode.

For the further analysis, we decide to use TDHF approach over other two for the following reasons. Firstly, sum rule approach is not well suited to account for a system that undergoes phase separation and moreover has to be provided with ground-state profiles from external theory. As for the hydrodynamic approach, it can be readily utilized for both statics and dynamics of considered system, however it is not obvious for which situation the system really behaves hydrodynamically. On the other hand, atomic orbital approach does not differentiate between collisionless and collisionally hydrodynamic regimes, as it should work well in both of them. The connection between hydrodynamic and atomic orbital methods will be further elaborated in Sec. 3.

To analyze the excitations, we will monitor the time evolution of the clouds' widths,

$$z(t) = \sqrt{\frac{\int \mathrm{d}^3\boldsymbol{r} \; r^2 n(\boldsymbol{r}, t)}{\int \mathrm{d}^3\boldsymbol{r} \; n(\boldsymbol{r}, t)}}. \tag{12}$$

To extract the frequencies, we either fit it (if possible) to the finite sum of sines with weights $\alpha_i$,

$$z(t) = \sum_i \alpha_i \sin(2\pi\omega_i t), \tag{13}$$

which is depicted in Fig. 3 in the form of markers, or we introduce adequately normalized Fourier transform of the evolution:

$$\tilde{\mathcal{F}}_\omega(\omega, k_F a) = \left| \int \mathrm{d}t \; e^{i2\pi\omega t} z(t) \right|^{1/3}, \tag{14}$$

$$\mathcal{F}_\omega(\omega, k_F a) = \frac{\tilde{\mathcal{F}}_\omega(\omega, k_F a)}{max_{\omega, k_F a = const.} \tilde{\mathcal{F}}_\omega(\omega, k_F a)}, \tag{15}$$

where the power 1/3 is introduced to improve visibility and the Fourier transform is normalized independently for every interaction strength $k_F a$ to have maximum at 1. It is plotted in the backgrounds of panels in Fig. 3 to guide an eye.

**In-phase oscillations**  First, we analyze radial oscillations with fully renormalized interaction for both weakly and strongly interacting gas. Fig. 3 (left, top) provides the results. Inclusion of the higher order terms in the interaction shifts the phase transition towards smaller values of $k_F a$ and increases the maximum value of the oscillation frequency, which is situated just before the transition occurs. It is now ca. $\omega/\omega_0 \approx 2.2$ in comparison to $\omega/\omega_0 \approx 2.1$ for the bare mean-field model.

After the transition, again there is only one dominating frequency that decreases with growing interaction and decreasing overlap of the clouds, achieving noninteracting value $\omega/\omega_0 \approx 2.0$ for a very strong repulsion. It suggests that in the fully separated regime, the domain wall has no effect on the frequency of the radial oscillations as compared to the non-interacting, polarized gas of $N$ fermionic atoms. This is the only case in which the oscillations can be fitted into a sum of sines for the interaction above the critical value. For all the other cases, the analysis is based only on the Fourier transform for each interaction strength.

**Out-of-phase oscillations**  Additionally, we consider out-of-phase radial oscillations. It is achieved by perturbing both clouds differently – when one of them undergoes squeezing of the trap, the other feels the trap to be widening. To have clean out-of-phase oscillations, small deviations of the densities ought to behave like $\delta n_+ = -\delta n_-$. However, it is nontrivial how to achieve such deviations as such a linear behavior needs infinitesimal perturbation and as a result, usual experimental setting mixes both in- and out-of-phase types of excitations. A potential experimental realization of such a scenario could involve imposing magnetic trapping on top of the optical one. The magnetic field would repel one of the species from the center of the trap, and attract the other one. Moreover, it is not unthinkable to find such an out-of-phase contribution in some other excitation schemes (e.g. quenching from one value interaction to another).

The results are presented in Fig. 3 (right, top). Below the critical value of the interaction, two branches are clearly visible. The upper one constitutes a contribution from in-phase oscillations that are excited by this particular scheme. The lower branch consists of out-of-phase contribution. Contrary to the upper one, the frequency decreases with the interaction, achieving ca. $\omega/\omega_0 \approx 1.2$ just before the phase transition. The Fourier transform shown as a background confirms that the lower branch is much stronger, with only small part of oscillations excited on the upper one.

After the transition, oscillations tend not to have a clear decomposition into small finite number of frequencies. However, the Fourier transforms of each of the time evolutions show strong contributions from the frequencies that can be considered extensions of two branches from the paramagnetic phase. The upper branch follows the trend from the in-phase perturbation scheme, decreasing and achieving noninteracting value. The same thing happens for the lower branch – it increases and goes to $\omega/\omega_0 \approx 2.0$ for very strong interaction. Contrary to the

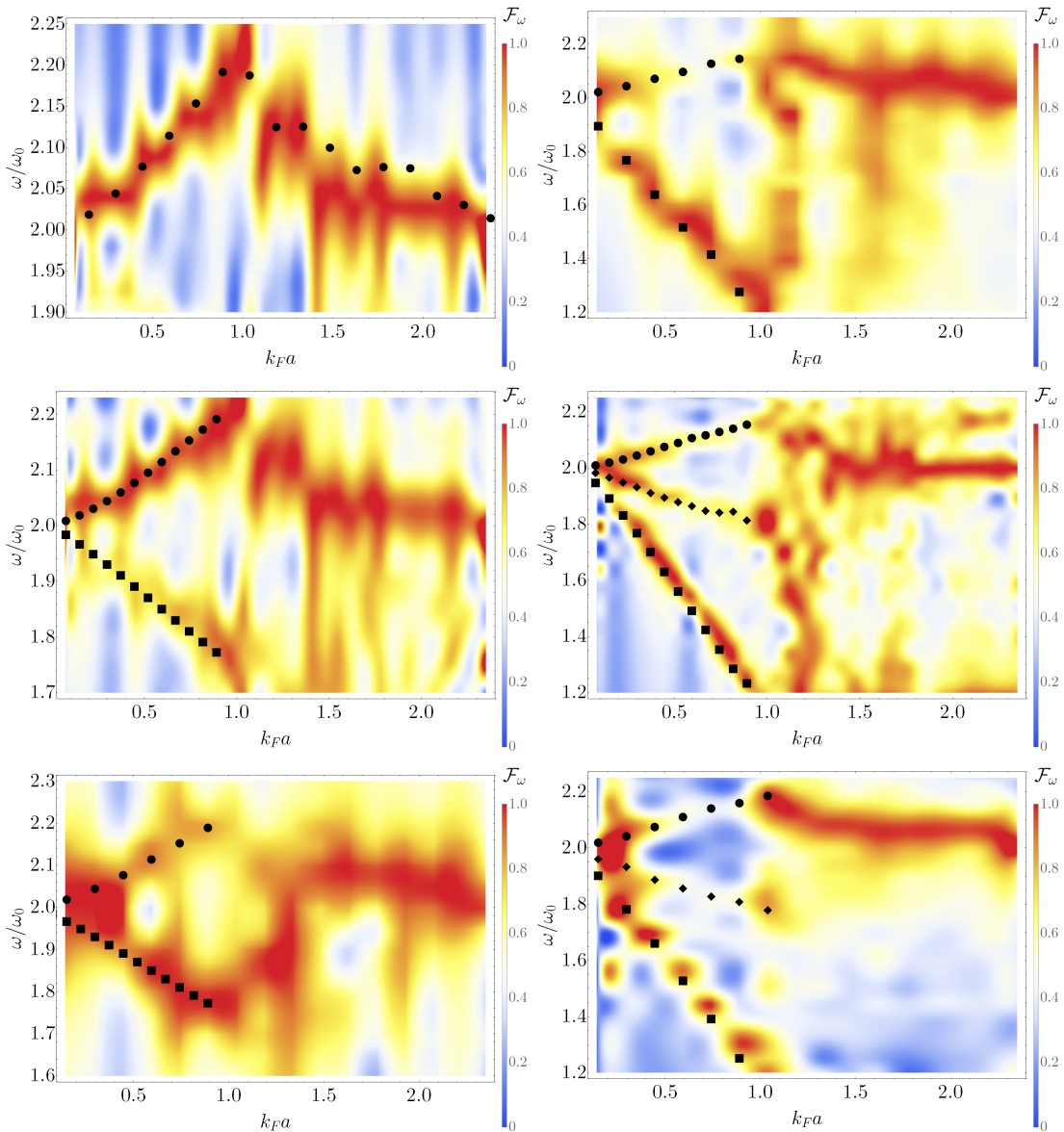

Figure 3: Frequencies of monopole (top row), radial compression (middle row) and radial quadrupole (bottom row) modes for in- (left column) and out-of-phase (right column) perturbations. The critical interaction for which the phase separation occurs reads $k_F a \approx 0.9$. In each of panels we present two results. First, filled markers (circles, squares, triangles) are associated with the frequencies that can be recovered from the fit to a sum of sines. Such a fit cannot be meaningfully performed for every interaction regime, especially after the transitions, therefore we additionally present Fourier transform of the oscillation run for each of the interaction values (for the definition see Eq. 14).

pre-transition regime, upper branch appears to be excited stronger. Additionally, contributions from other, lower- and higher-lying frequencies are clearly visible.

## 2.3 Radial modes

The next excitation that we analyze is a radial compression (breathing) mode. The perturbing scheme involves slightly compressing the gas in two out of three directions, leaving one of the

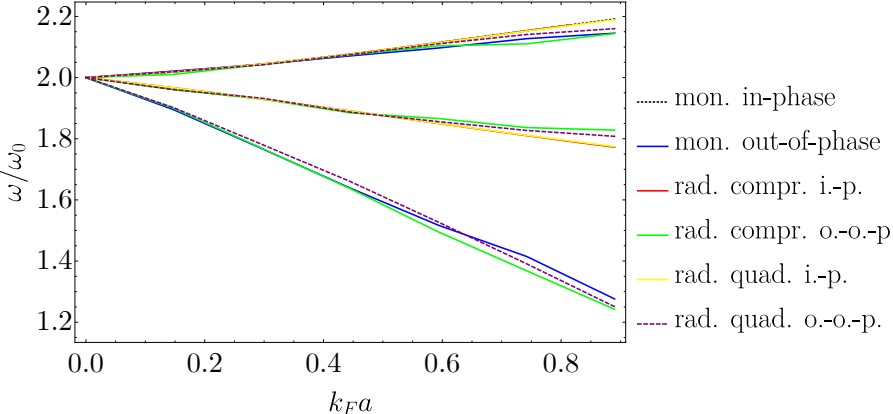

Figure 4: Comparison of all the frequencies in the fully overlapping phase. Combining all the results together, one can see that there are only three distinctive branches of frequencies that are excited during the perturbation schemes we have considered.

axes unperturbed. Again, we perform in- (Fig. 3 (left, middle)) and out-of-phase (Fig. 3 (right, middle)) calculations.

As for the in-phase case, we again can distinguish between two branches before the transition, however this time they are equally strong. The upper branch again achieves $\omega/\omega_0 \approx 2.2$ near the transition, but the lower one falls down to only $\omega/\omega_0 \approx 1.8$, giving rise to another distinctive excitation. After the transition, upper branch appears to be clearly more dominating, falling to noninteracting value for strongly interacting gas.

The out-of-phase case exhibits three distinctive branches – the upper ($\omega/\omega_0 \approx 2.2$ near the transition), the middle ($\omega/\omega_0 \approx 1.8$) and the lower ($\omega/\omega_0 \approx 1.2$) one. The latter clearly dominates. Again, after the transition, each of them is visible, going back to noninteracting regime, with the upper one staying the most pronounced.

The last excitation we consider is a type of a surface mode, called a radial quadrupole mode, in which gas is again excited only in two axes, however alternately, giving rise only to change of shape of the clouds, but not their volume. The results for in-phase perturbing scheme are presented in Fig. 3 (left, bottom) and for the out-of-phase one in Fig. 3 (right, bottom).

The in-phase excitation shows two distinctive branches – the usual upper one, and the lower one that near the transition achieves $\omega/\omega_0 \approx 1.8$. Contrary to previous cases, the lower branch stays stronger than the upper one after the transition. The out-of-phase case is characterized by the same previous three branches, with the lowest being the most dominating. This time, the upper branch stays the most pronounced over the transition.

One can immediately see by a direct comparison (see Fig. 4) that the branches (upper, middle and lower) are the same for different excitation schemes. However, each of the perturbations pronounces different branch, with the lowest one present only in out-of-phase settings.

## 3    Comparison to hydrodynamic approach

We now proceed to briefly compare the atomic orbital approach and the hydrodynamic (in this case sometimes called time-dependent Thomas-Fermi [68]) one. The most important assumption that underlies a hydrodynamic behavior is a fast relaxation of any dynamical distortions in momentum distribution towards a local Fermi sphere centered around local hydrodynamic

momentum of the collective flow. Such a thermalization is expected when both clouds overlap and interact by two-body collisions, but is not present in collisionless, noninteracting regime that governs the nonoverlapping parts of the clouds [2]. However, previous studies showed that such a hydrodynamic description works well for a spin-dipole mode for which experimental results are retrieved even in a weakly interacting regime. It is partially due to the fact, that the noninteracting frequency of this particular oscillation coincides with the hydrodynamic one. Then, corrections due to the interaction (even the weak one) seem to stem from the overlapping parts of the clouds, and can be described by the hydrodynamics. It is not obvious for which type of oscillations considered in this work hydrodynamic description can be used. We try to answer this question by comparing TDHF and hydrodynamic results.

The starting point for hydrodynamic methods is the density functional approach introduced for such a repulsive mixture in Ref. [59]. In this theory, the full Hamiltonian is given by $H = T_{\text{tot}} + E_{\text{int}} + E_{\text{pot}}$, and each of the terms can be written as a functional of one-particle density, making use of local density approximation. The first term, the total kinetic energy $T_{\text{tot}} = T + T_{\text{c}}$ consists of the intrinsic kinetic energy $T$, which is approximated by the Thomas-Fermi functional, $T = \sum_{j=\pm} \int \mathrm{d}\mathbf{r} \, \frac{3}{5} A n_j^{5/3}$ , and the kinetic energy of the collective motion, $T_{\text{c}} = \sum_{j=\pm} \int \mathrm{d}\mathbf{r} \, \frac{m}{2} n_j \mathbf{v}_j^2$. The potential energy is takes the form $E_{\text{pot}} = \sum_{j=\pm} \int \mathrm{d}\mathbf{r} \, V_j n_j$. The contact interaction term is given as an overlap between the density profiles of the components, $E_{\text{int}} = g \int \mathrm{d}\mathbf{r} \, n_+ n_-$.

To connect this DFT description to hydrodynamics, classical pseudo-wavefunction in the spirit of early Madelung's works [70] is introduced:

$$\psi = \begin{pmatrix} \psi_+ \\ \psi_- \end{pmatrix} = \begin{pmatrix} \sqrt{n_+} \, e^{i\frac{m}{\hbar}\chi_+} \\ \sqrt{n_-} \, e^{i\frac{m}{\hbar}\chi_-} \end{pmatrix}, \tag{16}$$

where $n_+ + n_- = \psi^\dagger \psi$ retrieves one-particle density, and $\nabla \chi_\pm = \mathbf{v}_\pm$ are the irrotational velocity fields of the collective motion. The Euler-Lagrange equation for these fields under the Hamiltonian $H$ then take hydrodynamic form:

$$\partial_t n_\pm = -\nabla(n_\pm \mathbf{v}_\pm),$$
$$m\partial_t \mathbf{v}_\pm = -\nabla \left( \frac{\delta T}{\delta n_\pm} + \frac{m}{2} \mathbf{v}_\pm^2 + V_\pm + g \, n_\mp \right). \tag{17}$$

To obtain frequency of small oscillations one can either linearize these equations or perform real-time evolution of the pseudo-Schrödinger equation that appears as a consequence of the inverse Madelung transformation:

$$i\hbar \partial_t \psi_\pm = \Big[ -\frac{\hbar^2}{2m} \nabla^2 + \frac{1}{2} \frac{\hbar^2}{2m} \frac{\nabla^2 |\psi_\pm|}{|\psi_\pm|}$$
$$+ A |\psi_\pm|^{4/3} + V_{tr} + g |\psi_\mp|^2 \Big] \psi_\pm. \tag{18}$$

Both methods generally yield the same results, and we use either of them, depending on the difficulty of performing the calculations in a given case.

We confirm that in the case of spherically symmetric trap hydrodynamics retrieves the TDHF results for radial monopole in both, in- and out-of-phase, cases. This statement is true for both types of interaction, renormalized and not renormalized. In the case of other modes, three branches were characterized – the upper (growing from $\omega/\omega_0 = 2.0$ to $\omega/\omega_0 \approx 2.2$), the middle (decreasing from $\omega/\omega_0 = 2.0$ to $\omega/\omega_0 \approx 1.8$), and the lower one (decreasing from $\omega/\omega_0 = 2.0$ to $\omega/\omega_0 \approx 1.2$). The first and the last one are retrieved quantitatively with TDHD

---

[2]However, some works suggested that Pauli principle can act as a substitute to interactions, and the hydrodynamic description can be working even for very weakly interacting Fermi gas [69].

approach in all cases. However, the middle branch is always characterized by $\omega/\omega_0 \approx 1.41$, independent of the interaction. It is not surprising, as such a solution of linearized hydrodynamic equations (in basic form, without renormalization) is explicitly independent from the coupling constant, yielding $\omega/\omega_0 = \sqrt{2}$.

It is important to stress the role of the particular choice of the external potential, namely spherically symmetric trap. As stated before, the dynamics is recovered accurately in cases where the hydrodynamic noninteracting frequency coincides with the true noninteracting one, namely $\omega/\omega_0 = 2$. For the spherically symmetric trapping, it occurs in the case of monopole and radial compression modes, but it does not for the radial quadrupole one. For different trappings, e.g. elongated traps, matching the noninteracting value is harder, as the hydrodynamic frequencies depend on the geometry. One example of such a scenario is a spin-dipole mode that equals $\omega/\omega_0 = 1$ and coincides for both hydrodynamics and noninteracting gas.

The results suggest that overlapping regions of fermionic clouds can be described hydrodynamically. It shows that if the hydrodynamic excitation frequency of the noninteracting gas is as it should be (coincidentally, as noninteracting gas is not hydrodynamic), the corrections due to interaction can be evaluated by the means of hydrodynamic description. It proves to be useful, when the TDHF description becomes intractable – namely when there are too many atoms in the system. Therefore, TDHF and TDHD seem to be complementary methods for such a use, TDHF being suitable for small systems, and TDHD being utilized to consider settings closer to experimental setups.

## 4 Conclusions

Recapitulating, we have analyzed small oscillations of repulsively interacting Fermi-Fermi mixture. Going beyond previous considerations, we investigated the gas on the repulsive energy branch, in contrast to attractive one of superfluid dimers. We did not limit ourselves to the paramagnetic phase in which the density profiles of each species are equal, but also covered the phase-separated state. We calculated monopole and two symmetry-breaking modes by the means of time-dependent Hartree-Fock methods. Moreover, we compared these results to the hydrodynamic approach, validating the regime of applicability of the latter to investigation of the dynamics of a weakly interacting Fermi gas. As a future line of work, in the presence of ongoing experiments on quantum mixtures, further investigation of validity of hydrodynamic formalism in such settings can be proposed.

**Funding information** The work was supported by (Polish) National Science Center Grant 2018/29/B/ST2/01308. Center for Theoretical Physics of the Polish Academy of Sciences is a member of the National Laboratory of Atomic, Molecular and Optical Physics (KL FAMO).

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
