# Peer review of "Collective oscillations of a two-component Fermi gas on the repulsive branch"

_SciPost Physics, doi:SciPost Phys. 8, 066 (2020)_

## Round 2 · Referee Report · Anonymous (Referee 1) · 2019-12-11

Strengths

1) The manuscript contains new physics: the authors investigate the frequencies of collective oscillations of two-component Fermi gas, for repulsive interactions.

Weaknesses

1) Some points are unclear.

2) The quality of Figs. 3-8 is poor.

Report

In this paper the authors presents a theoretical study of the frequencies of collective oscillations of two-component Fermi gas, on the repulsive branch of its energy spectrum. The analysis is carried out by means of some approximation, and for a limited number of particles (2 x 56), well below the typical numbers available in the experiments. Notwithstanding, the results may be interesting for the community working on this subject.

Before I can make a final recommendation, the author should consider the following points.

1) In equation (1) the authors employ a notation where the coordinates x_i refer to "both spatial and spin variables", but soon after they change to a different notation, with explicit indices for spin variables, and with an explicit dependence on time. I would suggest to unify the notation, for the sake of clarity.

2) The Fourier transform, which enters the discussion from pag. 7 on, it is never defined. Please include an explicit definition (in terms of formulas, I mean).

3) Figs. 3-8 have two problems, at least. They use much space, and this result in a dispersion of information. I would suggest to collect them in a single figure, with different panels, restricting the horizontal axis up to k_F*a=1 (nothing is plotted beyond that limit). Then, what are those blueish in Figs. 3,6,8? maybe the FT? It is totally unclear and aesthetically poorly plotted. Please revise the figure carefully.

4) The sentence "Recapitulating, we have filled the gap in the literature" (in the conclusions) is excessively presumptuous, I would suggest to remove it.

5) The English needs some revision. In particular, please check carefully the use of articles.

Requested changes

1) Unify the notation of equations (1)-(3).

2) Define qualitatively the Fourier transform.

3) Revise Figs. 3-8 (see report).

4) Avoid presumptuous claims.

5) Revise the English.

  • validity: ok
  • significance: ok
  • originality: good
  • clarity: low
  • formatting: good
  • grammar: acceptable

Author:  Piotr T. Grochowski  on 2020-03-31  [id 783]

(in reply to Report 1 on 2019-12-11)

We thank the Referee for the favourable report and insightful remarks. We believe that with a revised manuscript we satisfactorily answer the issues raised in the report.

  1. We unified the notation, changing the many-body wave function to express the spatial orbitals explicitly.

  2. We included explicit definitions of quantities we plot in the backgrounds of different panels in Fig. 3.

  3. We revised Figs. 3-8: We compiled them into one figure, added full legends and interpolated the Fourier transforms data to have a smooth background. However, we did not decide to remove data from the phase-separated regime as we believe it presents physically valuable results.

  4. We removed said sentence and checked for additional presumptuous statements in the text.

  5. We improved the language throughout the text.

---

## Round 2 · Referee Report · Anonymous (Referee 2) · 2020-2-24

Strengths

1-clear and well written article 2- TImely work 3- comparison with hydrodynamics and sum rule approaches

Weaknesses

1- choice of the trap geometry leading to misleading conclusion about comparison between hydrodynamics and TDHF

Report

In this article, the authors study the dynamics of a repulsive mixtures of fermionic gases confined in an isotropic trap within Hartree-Fock’s approximation. They focus on the lowest quadrupolar modes and consider identical and opposite excitations on the two species. The results are then compared to hydrodynamic approaches.
This article is interesting, clear and well written and the reported results are timely since the experimental study of repulsive Fermi gases is currently under way and is still largely open.
One shortcoming of the work though is the choice of a highly degenerate trap geometry that may lead to misleading conclusions if one wants to transpose the results of this work to more relevant trapping geometries (usually traps are anisotropic and more often than not strongly elongated). Indeed, one of the conclusions of the authors is that TDHF agrees with hydrodynamics (except for the quadrupole). This agreement is in my opinion coincidental and purely a consequence of the choice of trap. Indeed, if one considers an elongated trap, the prediction for the quadrupole modes of a weakly interacting gas is essentially twice the trapping frequencies. By contrast, hydrodynamics for instance predicts for the axial breathing mode a frequency equal to (12/5)^(1/2) times the axial trapping frequency. Conversely, is there a deep reason why hydrodynamics should be applicable to a weakly interacting gas in an isotropic trap?
Moreover, it would be interesting to compare the oscillation frequencies of the different modes, since it seems to me that the upper branches of all graphs are essentially all the same (this is pointed out when discussing symmetric vs antisymmetric excitation of the monopole compression mode, but I didn’t see it stated for the other modes). I have the impression that there are only three independent frequencies. Can the authors confirm? And if so, is there a simple explanation?
Finally, the numerical resolution of 3D nonlinear PDE being notoriously challenging, can the authors provide more details about the implementation of their numerical scheme (grid size, solver…)?

Requested changes

1-clarify the role of the trap geometry 2- Explicitely compare the oscillation frequencies 3- Provide more details on numerical implementation.

  • validity: high
  • significance: good
  • originality: good
  • clarity: top
  • formatting: excellent
  • grammar: excellent

Author:  Piotr T. Grochowski  on 2020-03-31  [id 784]

(in reply to Report 2 on 2020-02-24)

We thank the Referee for the positive report and valuable comments. We incorporated requested changes into the manuscript.

  • We agree that we did not include clear enough statements about the applicability of hydrodynamics and specifically, we did not stress the role of the trap geometry enough. First of all, we added a paragraph (a penultimate one) that follows on that issue:
It is important to stress the role of the particular choice of the external potential, namely spherically symmetric trap. As stated before, the dynamics is recovered accurately in cases where the hydrodynamic noninteracting frequency coincides with the true noninteracting one, namely $\omega / \omega_0 =2$. For the spherically symmetric trapping, it occurs in the case of monopole and radial compression modes, but it does not for the radial quadrupole one. For different trappings, e.g. elongated traps, matching the noninteracting value is harder, as the hydrodynamic frequencies depend on the geometry. One example of such a scenario is a spin-dipole mode that equals $\omega / \omega_0 =1$ and coincides for both hydrodynamics and noninteracting gas.

We believe that with introductory paragraph in that section, describing the usual applicability of hydrodynamics and its problems with noninteracting gas and with the last paragraph that explicitly comments on regimes in which we can use the framework:

It shows that if the hydrodynamic excitation frequency of the noninteracting gas is as it should be (coincidentally, as noninteracting gas is not hydrodynamic), the corrections due to interaction can be evaluated by means of hydrodynamic description.

we state the regime of applicability of the method clearly enough.

  • We added a plot in which we directly compare frequency branches in different cases, showing that there are only three prototypical collective excitations under the perturbation schemes we checked.

  • At the end of the introductory paragraph of Section 2, we added a paragraph with details of numerics.

---

## Round 3 · List of Changes

1. We unified the notation, changing the many-body wave function to express the spatial orbitals explicitly.
  2. We included explicit definitions of quantities we plot in the backgrounds of different panels in Fig. 3.
  3. We revised Figs. 3-8.
  4. We removed presumptuous statements in the text.
  5. We improved the language throughout the text.
  6. We elaborated on the applicability of hydrodynamics.
  7. We added a plot in which we directly compare frequency branches in different cases.
  8. We added a paragraph with details of numerics.

---

## Editorial Decision

published